# A framework for studying behavioral evolution by reconstructing ancestral repertoires

Damián G Hernández[1,2†], Catalina Rivera[1†], Jessica Cande[3], Baohua Zhou[1,4], David L Stern[3], Gordon J Berman[1,5]*

[1]Department of Physics, Emory University, Atlanta, United States; [2]Department of Medical Physics, Centro Atómico Bariloche and Instituto Balseiro, Bariloche, Argentina; [3]Janelia Research Campus, Howard Hughes Medical Institute, Ashburn, United States; [4]Department of Molecular, Cellular and Developmental Biology, Yale University, New Haven, United States; [5]Department of Biology, Emory University, Atlanta, United States

**Abstract** Although different animal species often exhibit extensive variation in many behaviors, typically scientists examine one or a small number of behaviors in any single study. Here, we propose a new framework to simultaneously study the evolution of many behaviors. We measured the behavioral repertoire of individuals from six species of fruit flies using unsupervised techniques and identified all stereotyped movements exhibited by each species. We then fit a Generalized Linear Mixed Model to estimate the intra- and inter-species behavioral covariances, and, by using the known phylogenetic relationships among species, we estimated the (unobserved) behaviors exhibited by ancestral species. We found that much of intra-specific behavioral variation has a similar covariance structure to previously described long-time scale variation in an individual's behavior, suggesting that much of the measured variation between individuals of a single species in our assay reflects differences in the status of neural networks, rather than genetic or developmental differences between individuals. We then propose a method to identify groups of behaviors that appear to have evolved in a correlated manner, illustrating how sets of behaviors, rather than individual behaviors, likely evolved. Our approach provides a new framework for identifying co-evolving behaviors and may provide new opportunities to study the mechanistic basis of behavioral evolution.

**\*For correspondence:**
gordon.berman@emory.edu

[†]These authors contributed equally to this work

## Introduction

Behavior is one of the most variable and rapidly evolving phenotypes, with notable differences even between closely related species (*Lorenz, 1958*; *Martins, 1996*). Variable behaviors and rapid behavioral evolution likely facilitates adaptation to new or varying environments and speciation (*Baier and Hoekstra, 1914*; *West-Eberhard, 2003*). Despite the importance of animal behavior, progress in revealing the genetic basis of behavioral evolution has been slow (*Gleason and Ritchie, 2004*; *Yamamoto and Ishikawa, 2013*; *Ellison et al., 2011*; *Shaw and Lesnick, 2009*). In contrast, recent decades have seen significant progress in understanding the genetic causes of morphological evolution (*Williams and Carroll, 2009*; *Shubin et al., 2009*; *Levine and Davidson, 2005*; *Stern and Frankel, 2013*).

While there are many potential reasons for the discrepancy between studies of behavioral and morphological evolution, including the lack of a fossil record for behavior, a key difficulty has been identifying which aspects of an animal's development and physiology are the proximate causes of behavior evolution. Evolutionary changes in behavior could emerge from alterations in the

developmental patterning of neural circuits (e.g., brain networks, descending commands, central pattern generators), changes in hormonal regulation that influence neural activity, or even from changes in non-neuronal morphology (*Baker et al., 2001*; *Massey et al., 2019*). Each of these possibilities could result in behavioral effects at different, yet overlapping, timescales – from muscle twitches to stereotyped suites of behaviors to longer-lived states like foraging or courtship or aging that may control the relative frequency of a given behavior. This complexity may make it difficult to identify the precise aspects of behavior that have evolved.

To address these difficulties, the standard approach in the genetic study of behavioral evolution has been to identify focal behaviors that exhibit robust differences between species, such as courtship behavior in fruit flies (*Cande et al., 2012*; *Cande et al., 2014*; *Ding et al., 2019*) or burrow formation in deermice (*Weber et al., 2013*; *Hu and Hoekstra, 2017*). It has been possible to identify genomic regions that correlate with quantitative changes in focal behaviors. However, usually multiple genomic regions are identified, each containing many genes. Given the large number of putative genes involved, combined with the possibility of epistatic interactions between loci, identification of the contributions of individual genes to behavioral evolution has progressed slowly.

An alternative approach to focusing on single behaviors is to examine the full repertoire of movements that an animal performs. By identifying sets of behaviors that evolve together, as was recently performed for hand-tuned traits in a study of birds-of-paradise evolution (*Ligon et al., 2018*), it may be possible to identify regulators of these suites of behaviors. This approach has been thwarted by the challenge of robustly measuring multiple behavioral phenotypes simultaneously. Recent progress in the unsupervised identification of animal behaviors across length and time scales, however, has made this approach possible (*Berman, 2018*; *Brown and de Bivort, 2018*). In this study, we introduce a quantitative framework for studying the evolutionary dynamics of large suites of behavior. We have focused initially on fruit flies, which provide a convenient model for this problem – both because they exhibit a wide range of complex behaviors and because unsupervised approaches can be used to map all of the animal movements captured in video recordings (*Berman et al., 2014*; *Cande et al., 2018*; *Berman et al., 2016*).

We recorded movies of isolated male flies from six species in a nearly stimulus-deprived environment. Because we did not record flies experiencing social and other environmental cues, we did not observe many charismatic natural behaviors, such as courtship and aggression. Nevertheless, we found that the behaviors they performed, including walking and grooming, contain species-specific information. We thus hypothesized that our quantitative representations of behaviors could be studied in an evolutionary context. To infer the evolutionary trajectories of behavioral evolution, we estimated ancestral behavioral repertoires with a Generalized Linear Mixed Model (GLMM) approach (*Hadfield, 2010*), which builds upon Felsenstein's approach to reconstructing ancestral states (*Felsenstein, 1985*; *Felsenstein, 2005*; *Hadfield and Nakagawa, 2010*; *O'Meara, 2012*). Using these results, we develop a framework that allows us to model the behavioral traits that co-vary both within a species and along the phylogeny. We found that within-species variance has a similar structure to long-lasting internal states of the animal that we characterized previously, and that inter-species variance can capture how disparate behaviors may have evolved together. This latter finding points toward the presence of higher order behavioral traits that would not have been detected by studying individual behaviors in isolation and that may be amenable to further evolutionary and genetic analysis.

## Experiments and behavioral quantification

We captured video recordings of all behaviors performed by single flies isolated in a largely featureless environment for multiple individuals from six species of the *Drosophila melanogaster* species subgroup: *D. mauritiana*, *D. melanogaster*, *D. santomea*, *D. sechellia*, *D. simulans*, and *D. yakuba* (*Cande et al., 2018*). Although the animals could not jump or fly in these chambers and were not expected to exhibit social or feeding behaviors, the flies displayed a variety of complex behaviors, including locomotion and grooming. Each of these behaviors involves multiple body parts that move at varying time scales. The species studied here were chosen because their phylogenetic relationships are well understood (*Clark et al., 2007*; *Obbard et al., 2012*; *Chyb and Gompel, 2013*; *Seetharam and Stuart, 2013*) (summarized in the tree seen in Figure 3), and genetic tools are available for most of these species (*Stern et al., 2017*). Since a single strain represents a genomic 'snapshot' of each species, we assayed multiple individuals from each of multiple strains of each species

to attempt to capture species-specific differences, and not variation specific to particular strains (see Materials and methods). In total, we collected data from 561 flies, each measured for an hour at a sampling rate of 100 Hz.

While previous studies have identified differences in specific behaviors, such as courtship behavior, between these species (*Cande et al., 2012*; *Ding et al., 2019*; *Yamamoto and Ishikawa, 2013*; *Auer and Benton, 2016*), here we assayed the full repertoire of behaviors the flies performed in the arena, with the aim of identifying combinations of behaviors that may be evolving together. To measure this repertoire, we used a previously described behavior mapping method (*Berman et al., 2014*; *Cande et al., 2018*) that starts from raw video images and finds each animal's stereotyped movements in an unsupervised manner. The output of this method is a two-dimensional probability density function (PDF) that contains many peaks and valleys (*Figure 1A*), where each peak corresponds to a different stereotyped behavior (e.g., right wing grooming, proboscis extension, running, etc).

Briefly, to create the density plots, raw video images were rotationally and translationally aligned to create an egocentric frame for the fly. The transformed images were decomposed using Principal Components Analysis into a low-dimensional set of time series. For each of these postural mode time series, a Morlet wavelet transform was applied, obtaining a local spectrogram between 1 Hz and 50 Hz (the Nyquist frequency). After normalization, each point in time was mapped using t-SNE (*van der Maaten and Hinton, 2008*) into a two-dimensional plane. Finally, convolving these points with a two-dimensional gaussian and applying the watershed transform (*Meyer, 1994*), produced 134 different regions, each of these containing a single local maximum of probability density that corresponds to a particular stereotypical behavior. We integrate over this local region of the probability density to calculate the probability that a fly is performing this behavior at a random point in time. Thus, we can associate each fly with a 134-dimensional real-valued vector that represents the probability of the fly performing a certain stereotyped behavior at a given time during the hour-long experimental session. We will refer to this quantity as the animal's *behavioral vector*, $\vec{P}$.

The behavioral map averaged across all six species is shown in *Figure 1A* and displays a pattern of movements similar to those we found in previous work, where locomotion, idle/slow, anterior/posterior movements, etc. are segregated into different regions (*Berman et al., 2014*; *Cande et al., 2018*). Averaging across all individuals of each species, we found the mean behavioral vector for

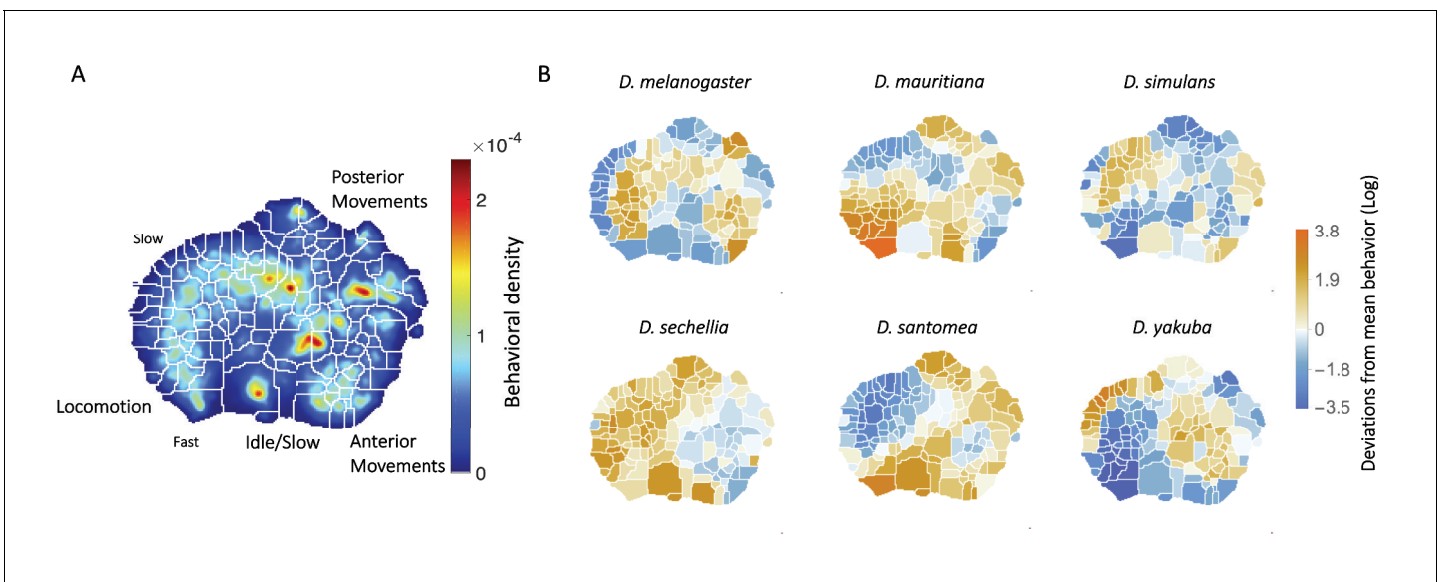

**Figure 1.** Behavioral repertoires of *Drosophila*. (**A**) The behavioral space probability density function, obtained using the unsupervised approach described in *Berman et al., 2014* on the entire data set of 561 individuals across all species. Coarse grained behaviors corresponding to the different types of movements exhibited in the map are shown as well. (**B**) The relative performance of each of the 134 stereotyped behaviors for each of the six species. Each region here represents a behavior, and the color scale indicates the logarithm of the fraction of time that each species performs the specified behavior divided by the average across all species.

each species (*Figure 1B*) and observed that each species performs certain behaviors with different probabilities. For example, *D. mauritiana* individuals spend more time performing fast locomotion than all other species on average, and *D. yakuba* individuals spend much of their time performing an almost species-unique type of slow locomotion, but little time running quickly.

These average probability maps provide some insight into potential species differences, but to identify species-specific behaviors, we also need to account for variation in the probability that individuals of each species perform each behavior. One way to address this problem is to ask whether an individual's species identity can be predicted solely from its multi-dimensional behavioral vector. To explore this question, we first used t-SNE to project all 561 individuals into a two-dimensional plane (*Figure 2A*), using the Jensen-Shannon divergence as the distance metric between individual behavioral vectors. In this plot, different colors represent different species, and different symbols with the same color represent different strains within the same species. Although species do not segment cleanly into separate regions of this plane, the distribution of species is far from random, with individuals from the same species tending to group near to one other. Given this structure, there is likely species-specific information in the behavioral vectors.

To quantify this observation, we applied a multinomial logistic regression classifier to the data, performing a six-way classification based solely on the high-dimensional behavioral vectors. After training, the classifier correctly classified $89 \pm .2\%$ of vectors in our test set (a randomly selected 30% of the entire data set that was not used during training). Moreover, the confusion matrix (*Figure 2B*) revealed no systematic misclassification bias amongst the species. Note that we have used a relatively simple classifier compared to modern deep learning methods (*Goodfellow et al., 2016*), so these results likely represent a lower bound on the distinguishability of the behavioral vectors. Thus, the behavioral vectors contain considerable species-specific information. We therefore proceeded to explore how these behavioral vectors may have evolved along the phylogeny.

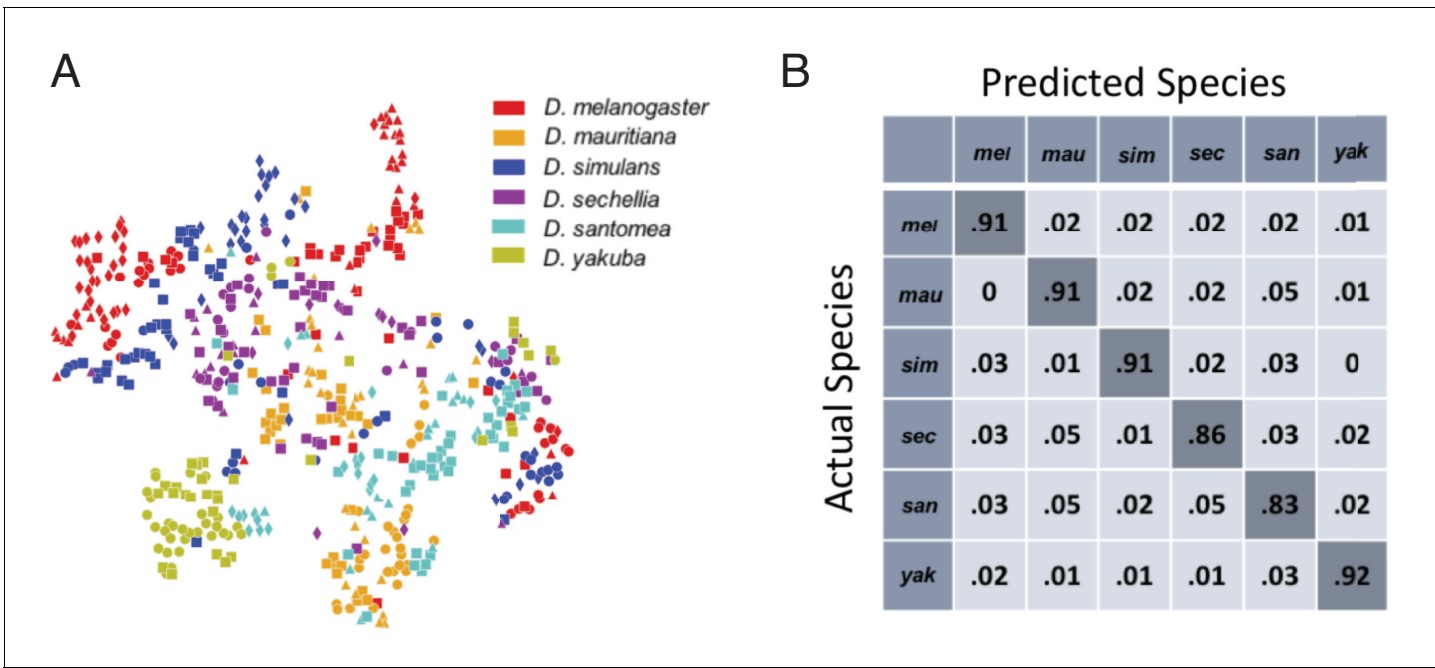

**Figure 2.** Classification of fly species based on behavioral repertoires. (**A**) A t-SNE embedding of the behavioral repertoires shows that behavioral repertoires contain some species-specific information. Each dot represents one individual fly, with different colors representing different species and different symbols with the same color representing different strains within the same species. The distance matrix (561 by 561) used to create the embedding is the Jensen-Shannon divergence between the behavioral densities of individual flies. (**B**) Confusion matrix for the logistic regression with each row normalized. All the values are averaged from 100 different trials. The standard error is less than 0.01 for the diagonal elements and less than 0.005 for each of the off-diagonal elements.

## Reconstructing ancestral behavioral repertoires

Multiple methods have been proposed for reconstructing ancestral states from data collected from extant species (*Felsenstein, 1985*; *Felsenstein, 2005*; *Yang, 2006*; *O'Meara, 2012*; *Royer-Carenzi and Didier, 2016*). These methods generally fall into two camps: parsimony reconstruction, which attempts to reconstruct evolutionary history with the fewest number of evolutionary changes (*Cunningham et al., 1998*), and diffusion-processes, which model evolution as a random walk on a multi-dimensional landscape (*Hadfield and Nakagawa, 2010*). Given the high-dimensional behavioral vectors that we are attempting to model, a diffusion process is more likely to capture the inter-trait correlations that we would like to understand. Thus, we focus on a diffusion-based model here.

Given a phylogeny for a collection of species, we modeled how species-specific complexes of behaviors might have emerged. We assumed that each animal's behavior is a quantitative trait with an additive random effect, that is, each animal's behavior is a trait that results from the additive effects of many genetic loci, each of small effect, that is combined with a non-genetic effect that represents inter-specific variation. We do not, however, assume that all behaviors evolve independently of each other. Thus, we are interested in predicting (1) whether intra- and inter-species variation can be separated to identify independently evolving sets or linear combinations of behaviors and (2) how behaviors co-vary along the phylogeny, potentially revealing co-evolving suites of behaviors.

We assumed that the observed flies' behaviors evolved via a diffusion process with Gaussian noise from a common ancestor along the known phylogenetic tree. Note that this is a less restrictive assumption than neutrality, as multiple traits under selection may evolve in a correlated manner. Specifically, we fit a multi-response Generalized Linear Mixed Model (GLMM) to the data, using the approach described in *Hadfield, 2010*, modeling the evolutionary process such that the logarithm of the *behavioral vector, $\vec{P}$*, for each individual ($\vec{l} = (l_1, ..., l_{K=134})$), is given by

$$\vec{l} = \vec{\mu} + \vec{\rho} + \vec{e}, \tag{1}$$

where $\vec{\mu}$ is the mean behavior of the common ancestor (treated as the fixed effects of this model), and $\vec{\rho}$ and $\vec{e}$ are the random effects corresponding to the phylogenetic and individual variability, respectively. We assume that these random effects are generated from the multi-dimensional normal distributions $\mathcal{N}(\vec{0}, A \otimes V^{(a)})$ (phylogenetic) and $\mathcal{N}(\vec{0}, I \otimes V^{(e)})$ (individual). Here, the matrix $A$ represents the information contained in the phylogenetic tree, with $A_{ij}$ being proportional to the length of the path from the most recent common ancestor of species $i$ and $j$ to the common ancestor. $A$ is normalized so that the diagonal elements are all equal to 1. Therefore, $A_{ij}$ represents the phylogenetic similarity between a pair of species. $I$ is the identity matrix, and $V^{(a)}$ and $V^{(e)}$ are the phylogenetic and within-species covariance matrices, respectively.

We fit $\mu$, $V^{(a)}$, and $V^{(e)}$ using Markov Chain Monte Carlo (MCMC) simulations, confirming that the MCMC converged using the Gelman-Rubin diagnostic (see Materials and methods, *Figure 3—figure supplement 1*). In addition, our model is able to infer the mean endpoint behavioral repertoires (*Figure 3—figure supplement 2*), providing confidence that our model is consistent with our input data. In addition to the inferred behavioral states corresponding to the common ancestor, $\bar{P}^{Anc}$, we also reconstructed the mean behavioral representations for the intermediate ancestors (*Figure 3*).

We also found that the model that allows behavioral co-evolution out-performs a model where each behavioral trait evolves independently. Specifically, we fit a model where behavioral correlations between individuals of different species were removed by enforcing that $V^{(a)}$ and $V^{(e)}$ must be diagonal matrices, a reduction of more than 17,500 parameters compared to the full model (see Materials and methods for details). The phylogenetic ancestral reconstruction was then made for each behavioral trait separately. To compare the relative performance of these models, we used the Deviance Information Criterion (DIC) (*Spiegelhalter et al., 2002*), a commonly used assessment tool for MCMC-fit hierarchical models that lack a good estimate for the number of effective parameters. Like other information-theoretic model selection criteria (e.g., the Akaike Information Criterion or the Bayesian Information Criterion), smaller values of the DIC imply a larger posterior probability of the model given the available data. Despite the large reduction in the number of parameters for the independent-trait model, the DIC for the independent-trait model was substantially higher ($\text{DIC} = (242 \pm 2) \times 10^3$) than the DIC for the full model ($\text{DIC} = (114 \pm 2) \times 10^3$). Moreover, the full model was able to predict the inter- and intra-species covariances between dissimilar pairs of

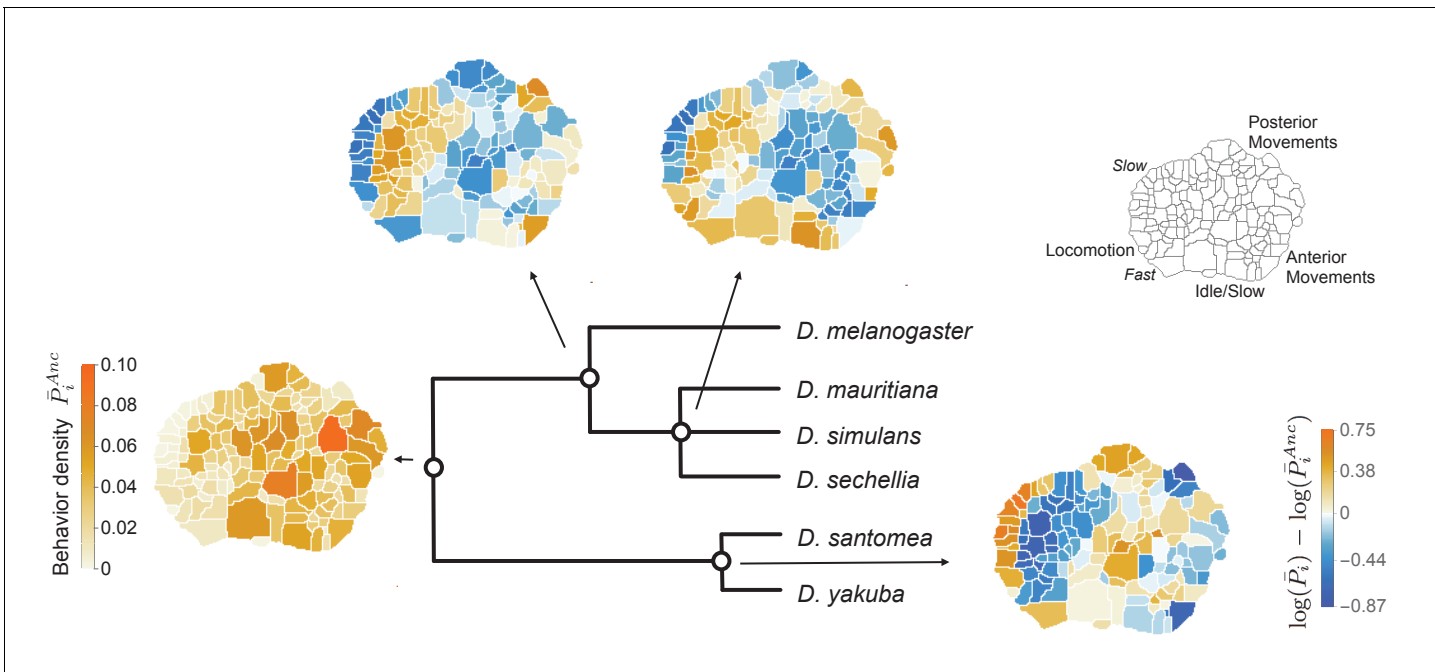

**Figure 3.** Reconstructed behavioral repertoires using the GLMM. Inferred probabilities of the behavioral traits for the ancestral states are plotted at the denoted locations along the phylogeny. Except for the common ancestor, ancestral states are plotted with respect to the closest ancestor. For each behavioral trait, $i$, in the intermediate ancestors, we show: $\log(\bar{P}_i) - \log(\bar{P}_i^{Anc})$, where $\bar{P}i$ and $\bar{P}_i^{Anc}$ correspond to the inferred mean behavioral trait for the given ancestor and its closest ancestor, respectively. Coarse grained behaviors corresponding to different types of movements are shown on the top right corner.

The online version of this article includes the following figure supplement(s) for figure 3:

**Figure supplement 1.** Gelman Rubin diagnostic for model parameters inferred using MCMC.
**Figure supplement 2.** Comparison between measured and inferred behaviors (on a log scale) for each of the extant species.
**Figure supplement 3.** Comparison of the independent focused trait approach vs the repertoire approach for a pair of behaviors.

behaviors (*Figure 3—figure supplement 3*). Hence, modeling the evolution of the full behavioral repertoire captures the structure of the observed data better than a single trait approach.

## Individual variability and long timescale correlations

While it is not possible to directly test the accuracy of our ancestral state reconstructions (*Figure 3*), the inferred covariance matrices generate predictions about behavioral and genetic correlations that are, in principle, testable. We therefore focus on the fitted covariance matrices, $V^{(e)}$ and $V^{(a)}$ (each in $\mathbb{R}^{134 \times 134}$), which account for within-species and phylogenetic random effects, respectively.

We will focus first on the intra-species covariance matrix, $V^{(e)}$. We note first that the matrix exhibits a modular structure (*Figure 4A*). After rearranging the behavior order via an information-based clustering procedure (*Slonim et al., 2005*), we see that a block diagonal pattern emerges, with positive correlations lying within the blocks and negative correlations lying off the diagonal. The details of this particular clustering approach are described in Materials and methods, but we find that the results are nearly identical for several different clustering methodologies (*Figure 4—figure supplement 1*). Quantifying the matrix's modularity via the average within-cluster dissimilarity,

$$\langle d \rangle = \sum_k p(C_k) \sum_{i,j \in C_k} \frac{1}{2} \left[ 1 - \frac{V_{ij}^{(e)}}{\sqrt{V_{ii}^{(e)} V_{jj}^{(e)}}} \right], \tag{2}$$

where $C_k$ is the set of all behaviors belonging to the $k$ th cluster, we find that $\langle d \rangle \approx 0.30$ and 0.22 for the 3- and 6-cluster solutions, respectively. These values are significantly smaller than the average distances obtained using random cluster assignments ($\langle d \rangle = 0.46 \pm 0.03$ and $0.45 \pm 0.04$ for 3 and 6

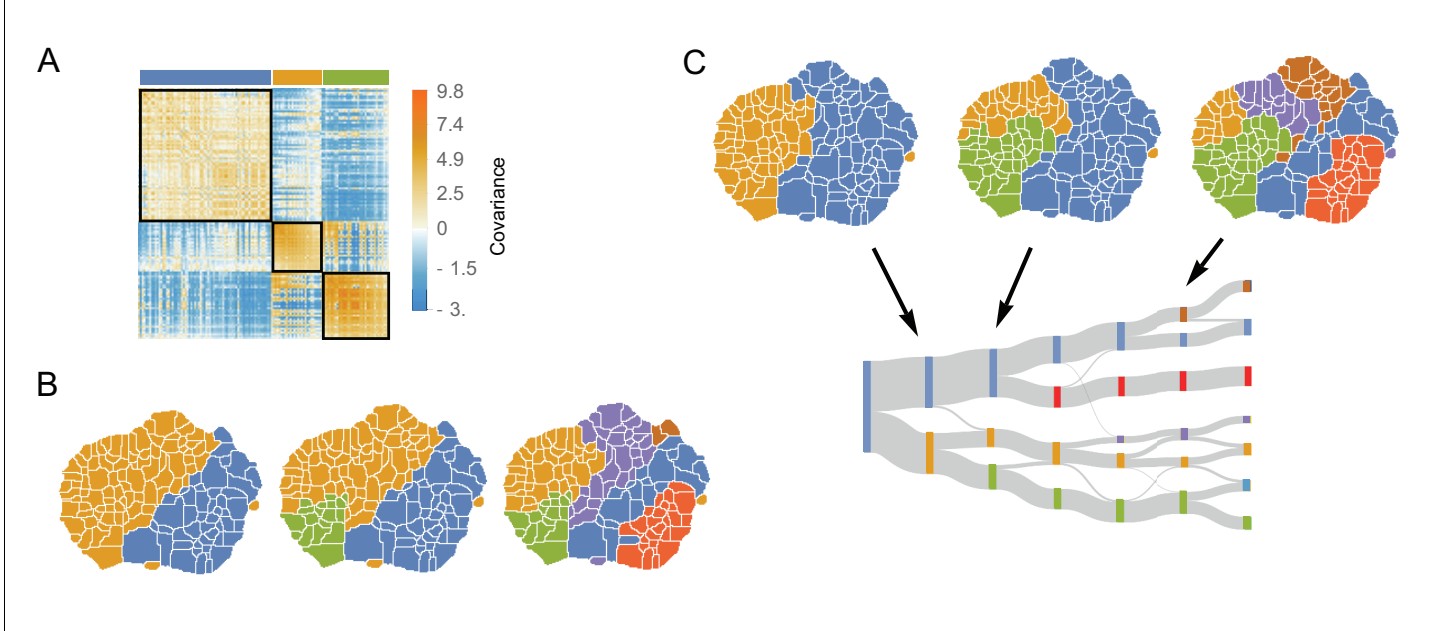

**Figure 4.** The structure of variability between flies of the same species relates to long timescale transitions in behavior. (A) The intra-species behavioral covariance matrix ($V^{(e)}$), with columns and rows ordered via an information-based clustering algorithm (*Slonim et al., 2005*). The black squares represent behaviors that are grouped together in the three-cluster solution. (B) Behavioral map representation of the clustering solutions. The two-, three-, and six-cluster solutions are shown on top (colors on the three cluster solution match those above the plot in A). The clusters are all spatially contiguous and break down hierarchically (see *Figure 4—figure supplement 1* for more examples). (C) Clustering structure of the behavioral space obtained finding the optimally predictive groups of behaviors (see text for details). Note how these clusterings are very similar to the clusterings in B, despite having been derived from an entirely independent measure.

The online version of this article includes the following figure supplement(s) for figure 4:

**Figure supplement 1.** Behaviors clustered according to the individual covariance matrix using three different clustering methods.

**Figure supplement 2.** Modularity of the intra-species behavioral covariance matrix using information based clustering.

**Figure supplement 3.** Coarse-grained behavioral representations that are optimally predictive of the future behavior states via DIB.

clusters respectively, see *Figure 4—figure supplement 2* for other numbers of clusters and clustering methods). Thus, we can conclude that the intra-specific covariance matrix has a far-from-random modular structure, implying that between individuals of the same species, groups of behaviors tend to vary together in a stereotyped manner.

Moreover, these groups of behaviors that co-vary together within a species are not random collections of behaviors. Instead, we found that co-varying clusters are spatially contiguous in the behavioral map, implying that covariances of groups of similar behaviors (behaviors involving moving similar parts of the animals' bodies at similar speeds) compose much of the observed intra-species variance. The clustering method does not take the spatial structure of the behavioral map into account at all (just the values in $V^{(e)}$), so the clusters of local behaviors in the behavior map reflect underlying similarity in the covariance of nearby behaviors, rather than an artifact of the algorithm. Moreover, co-varying clusters are hierarchically organized, where coarse-grained co-varying behaviors can be sub-divided into smaller co-varying clusters (*Figure 4B*), a feature that is not guaranteed by the information-based clustering algorithm.

This hierarchical structure of the behavioral map is reminiscent of the hierarchical temporal structure of behavior that was hypothesized originally by ethologists (*Tinbergen, 1951*; *Deutsch et al., 2020*) and was observed to optimally explain the history-dependent long timescale non-stationary structure of *Drosophila melanogaster* behavioral transitions (*Berman et al., 2016*). Thus, we hypothesized that the structure of the intra-species covariance matrix might be linked to deviations from statistical stationarity in the behavioral data that were not explicitly measured in the unsupervised clustering or modeled in the GLMM.

To explore this connection further, we performed an analysis that is analogous to the single-species study in *Berman et al., 2016*, finding coarse-grainings of the behavioral space (i.e., a description of the behavioral space using fewer behaviors) that are optimally predictive of the future behaviors that the flies perform. Specifically, if $b(t)$ is the behavior that a fly performs at time $t$, we would like to create a clustered version of our behavioral map, $Z$, such that we maximize the information that $z(t) \in Z$, the cluster that the fly is in at time $t$ contains as much information about the future behavior of the fly, $b(t + \tau)$ as possible. To keep $Z$ from separating each behavior into its own cluster, we also need to make sure that $Z$ is as simple a clustering as possible (i.e., a smaller number of clusters and a more even distribution of time spent within each clusters).

To be more precise, we calculated $Z$ using the the Deterministic Information Bottleneck (DIB) method (*Strouse and Schwab, 2017*). This approach minimizes the functional

$$\mathcal{J}_\tau = -I(Z(t); Z(b + \tau)) + \gamma \mathcal{H}(Z), \tag{3}$$

where $b(t + \tau)$ is a fly's behavior at time $t + \tau$, $Z(t)$ is the coarse-grained behavior visited at time $t$, $I(b(t); Z(t + \tau))$ is the mutual information between these quantities, $\gamma$ is a positive constant, and $\mathcal{H}(Z)$ is the entropy of the coarse-grained representation (see Materials and methods). As $\gamma$ is increased, progressively simpler, but less predictive, representations are found.

Applying this method to the data, pooled across all six species and using $\tau = 50$ (*Figure 4C*, *Figure 4—figure supplement 3*), we found the same hierarchical division of the behavioral map that was observed for freely moving *D. melanogaster* (*Berman et al., 2016*). Moreover, we found that the structure of the space using this approach closely mirrors the structure found via directly clustering the intra-species covariance matrix, $V^{(e)}$ (*Figure 4C*). Quantifying the similarity between both clustering partitions by calculating the Weighted Similarity Index (WSI), a modification of the Rand Index (*Rand, 1971*) (Materials and methods), the WSI between the information-based clustering method and the predictive information bottleneck for three clusters is $WSI = 0.73$, and $WSI = 0.87$ for six clusters. For random clusterings, we would expect to observe $0.51 \pm 0.02$ and $0.70 \pm 0.01$ for 3 and 6 clusters, respectively, indicating a non-random overlap between these two partitions. *Figure 4—figure supplement 1*, shows that this result is independent of the clustering method and the number of clusters.

The overlap between these two coarse-grainings indicates that most individual variability in the behaviors we observe results from non-stationarity in behavioral measurements, rather than from individual-specific variation. That is, much of the intraspecific variation appears to reflect flies recorded when they were experiencing different hidden behavioral states (e.g., circadian state, hunger, etc.), rather than reflecting fixed (environmental or genetic) differences between flies. This variation may have arisen because, although we controlled many variables (e.g., fly age, circadian cycle, temperature, and humidity), it is not possible to control for all internal factors (e.g., hunger, arousal, etc.) that affect an animal's behavioral patterns (*Anderson, 2016*). The temporal coarse-graining of the behavioral space that we found via the DIB provides insight into these non-stationarities, as they are optimally predictive of the fly's future behaviors. Given the contiguous nature of these regions, this result means that flies tended to stay within specific regions of the behavioral space much longer than one would assume from a Markov model, hinting that there is an important connection between variability across animals and variability between animals.

More precisely, these results imply that variation in behavior observed among individuals, especially in non-manipulated settings, may often reflect a large component of hidden behavioral states (*Figure 5A*). Thus, it may be possible to improve upon behavioral measurements in many settings by controlling for the variability associated with these hidden states. For example, just because one fly performs less anterior grooming than another may reflect that the animal is in a different long time-scale behavioral state, rather than that the animal has a genetically encoded preference for reduced grooming.

A potential method for accounting for these artifacts is to normalize each individual's behavioral density such that the amount of time that the animal spends in each of the coarse-grained regions is equalized. In other words, the amount of time spent anterior grooming, locomoting, etc. are set to be the same for all animals, thus accounting for the variability associated with the inferred hidden states. Mathematically, if $P_i$ is the probability of observing behavior $i$, and $C_i$ is the clustering assignment of this behavior, we can define a normalized probability, $\hat{P}_i$, via

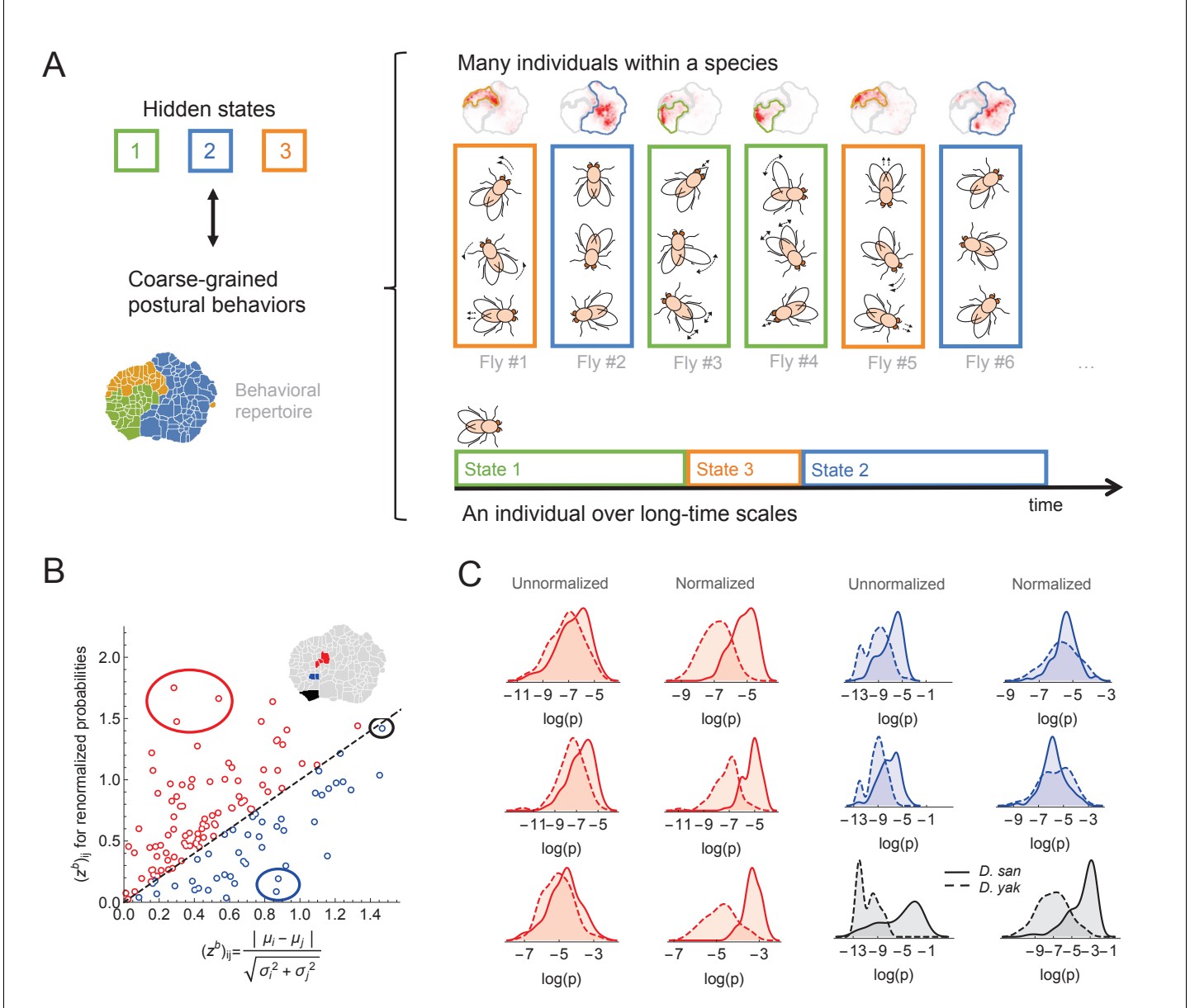

**Figure 5.** Variability within a species, long timescale transitions, and hidden states modulating behavior. (A) A cartoon of the hypothesized relation between individual variability within a species and long timescale transitions through hidden states. (B) Accounting for the long timescale dynamics - by adjusting for the amount of time spent in each coarse-grained region (here, the six cluster solution at the top right of *Figure 4C*) - affects the measured behavioral distributions between *D. santomea* and *D. yakuba*. Shown is the comparison of the Mahalanobis distance ($(z^b)_{ij}$) between behavioral distributions before (x-axis) and after (y-axis) adjusting. (C) Kernel density estimates of the distributions for the circled behaviors in (B) on the left before (left) and after (right) adjustments. Solid lines represent *D. santomea* and dashed lines represent *D. yakuba*.

$$\hat{P}_i = \frac{\bar{P}^{(C_i)}}{P_i^{(C_i)}} P_i, \tag{4}$$

where $P_i^{(C)} = \sum_{k \in C} P_k$ is the total density in cluster $C$ for an individual fly and $\bar{P}^{(C)}$ is the average across all animals.

We found that applying this normalization to our data often results in substantial changes in the inferred distributions of behavioral densities. For example, *Figure 5B* displays how the difference in behavioral density between *D. santomea* and *D. yakuba* (as measured by the Mahalanobis distance

between the distributions) alters as a result of normalization. For some behaviors, the signal increases (red points), and in some cases, it decreases (blue points). Thus, it is important to take these non-stationary effects into account when estimating how often single behaviors are performed in studies of behavioral evolution. To measure these non-stationary effects, many behaviors must be measured, not just a focal behavior, thus partially explaining the relative success of our multi-trait model compared to a model where each trait is analyzed independently.

## Identifying phylogenetically linked behaviors

One of the advantages of our approach is that we separate variations in behavior corresponding to evolutionary patterns, the phylogenetic variability, from variations among individuals of the same species. By studying the properties of the phylogenetic covariance matrix ($V^{(a)}$), we can identify multiple behaviors that may have evolved together.

We first characterized the coarse-grained structure within $V^{(a)}$ through the information-based clustering used in the previous section (*Slonim et al., 2005*) (see Materials and methods). As seen in *Figure 6A*, the phylogenetically co-varying clusters are not spatially contiguous in the behavioral map. This finding is in contrast to the spatial contiguity we observed for the intra-species covariance matrix (*Figure 4B*). For example, the two-cluster solution (*Figure 6A*, left) groups the behavioral space into side legs movements (middle of the behavioral map) and certain locomotion gaits (far left of the behavioral map) versus the rest of behaviors. Similarly, when the matrix is clustered into a

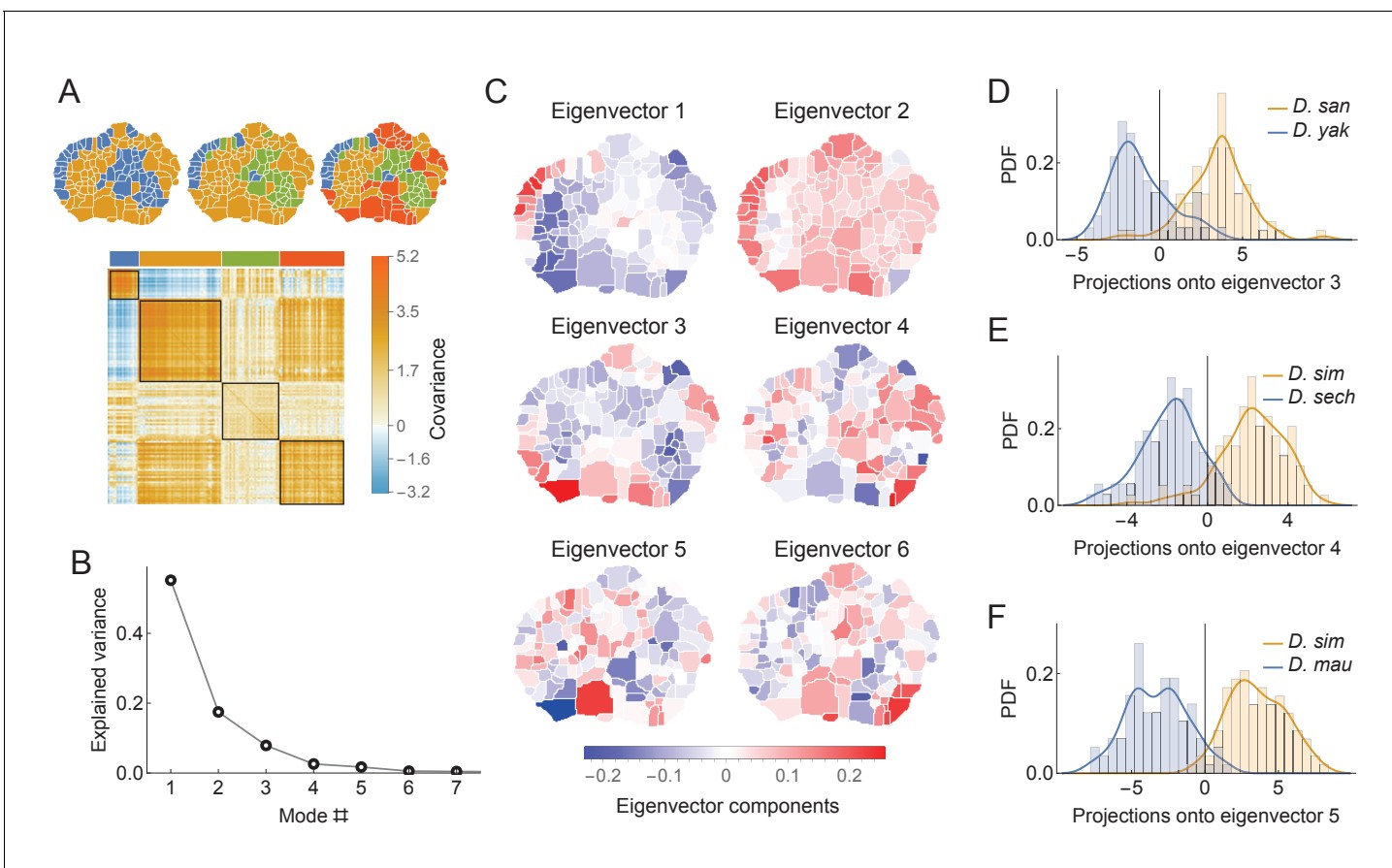

**Figure 6.** Phylogenetic variability and behavioral meta-traits. (A) (top) Clustering the phylogenetic covariance matrix (using the same information-based clustering method from *Figure 4*), we observe that the clusters are no longer spatially contiguous. (bottom) The phylogenetic covariance matrix reordered according to four clusters (colors corresponding to the four-cluster map above). (B) Fraction of variance explained by the largest eigenvalues of the phylogenetic covariance matrix. (C) The eigenvectors corresponding to the largest six eigenvalues. (D) Distributions of the projections of individual density vectors from *D. santomea* and *D. yakuba* onto eigenvector 3. (E) Same as in D but using projections of individuals from *D. sechellia* and *D. simulans* onto eigenvector 4. (F) Same as in D but using projections of individuals from *D. simulans* and *D. mauritiana* onto eigenvector 5.

larger number of clusters, correlated groups are not contiguously arranged within the behavior map. Thus, our model predicts that many non-similar behaviors are evolving in a correlated manner.

To quantify these patterns as traits, we decomposed $V^{(a)}$ via an eigendecomposition. As seen in *Figure 6B*, almost all of the variance within the matrix can be explained with only the first six eigenmodes. These eigenvectors (*Figure 6C*) share similar non-local structure to the clusterings described above. By projecting individual behavioral vectors onto these eigenvectors, the resulting dot products represent a meta-trait that is a linear combination of phylogenetically linked behaviors.

These evolving meta-traits may be suitable targets for further neurobiological or genetic studies. Three examples of these distributions are shown in *Figure 6D–F* for several pairs of closely related species. These three examples were not chosen at random, but instead because they showed significant differentiation between species. The aim of this analysis is not to show that all meta-traits would differ between all pairs of species, which is unlikely, but rather that it is possible to identify synthetic meta-traits that could be further interrogated with experimental methods.

## Discussion

We have developed a quantitative framework to study the evolution of behavioral repertoires, using fruit flies (*Drosophila*) as a model system. We started with observations of 561 individuals from six extant species behaving in an unremarkable environment. This assay did not include social behaviors, such as courtship and aggression, nor many foraging behaviors. Thus, at first glance, it might seem like we had excluded most species-specific behaviors from the analysis. Nonetheless, we found that other complex behaviors, like walking, running, and grooming, exhibit species-specific features that can be used to reliably assign individuals to the correct species. Thus, the motor patterns of behaviors that are not normally investigated for their species-specific features are likely evolving between even closely related species. It is not clear, however, if these differences reflect natural selection or genetic drift. All of these behaviors are critical to individual survival, however, so it is possible that these behaviors have evolved, at least in part, in response to natural selection. It is clear, however, that the underlying mechanisms, and perhaps the neural circuitry, controlling these behaviors must have evolved.

Inspired by these observations, we estimated patterns of behavioral evolution in the context of a well-understood phylogeny. We fit a Generalized Mixed Linear Model to our behavioral measurements and the given phylogeny to reconstruct ancestral behavioral repertoires and the intra- and inter-species covariance matrices. We found that the patterns of intra-species variability are similar to long timescale behavioral dynamics that violate statistical stationarity - a result we reported previously in a study of a single species (*Berman et al., 2016*). This result suggests that much of the intra-specific variability that emerged by sampling flies under well-controlled conditions reflects variability in the hidden behavioral states of individual flies. While it may be challenging to conceptualize that seemingly simple behaviors, like the pace of walking and running, are reflective of an underlying long time scale behavior state, many short time scale behaviors, such as the individual movements involved in grooming (*Seeds et al., 2014*), courtship (*Calhoun et al., 2019*; *Deutsch et al., 2020*) and aggression (*Hoopfer, 2016*; *Duistermars et al., 2018*) reflect behaviors performed only, or mainly, in the context of a longer lasting behavioral state. These types of long timescale variability may be a statistical confound for evolutionary and experimental studies of behavior. We therefore propose a method to control for these internal states by normalizing the frequency of behaviors relative to an estimate of an animal's non-stationary states. This method improved the accuracy of behavioral phenotyping and dramatically altered estimates of some species-specific behaviors. For more focused studies, it may not be necessary to measure the full suite of behaviors to effectively normalize for behavioral state, since state can sometimes be estimated from a smaller number of behaviors. In fact, targeted studies of charismatic behaviors, including behaviors associated with aggression or courtship, often implicitly normalize by behavioral state.

Given our estimates for how suites of behaviors evolved, we examined whether the inter-species covariance matrix could be used to identify behavioral meta-traits that might be subjected to further evolutionary and experimental analysis. We identified multiple suites of behaviors that differed between closely related species, providing a starting point for further analysis of how the mechanisms underlying these suites of behaviors have evolved.

There are multiple possible interpretations of these phylogenetically correlated behaviors. For instance, at the neural level, each of these groups of movements may reflect a motor response to shared upstream commands (*Cande et al., 2018*). Here, for example, different types of locomotion might be controlled through the same descending neural circuitry, but due to evolutionary changes, the same commands could lead to different behavioral outputs, as has been observed in fly courtship patterns (*Ding et al., 2019*). Alternatively, at the genetic level, multiple behaviors may be linked by pleiotropic effects of individual genetic changes. Finally, groups of co-evolving traits may not be linked mechanistically, but co-evolution may instead reflect selection on suites of behaviors. For example, the male neurons that drive fly courtship song production in the ventral nerve cord are unlikely to be related to the female neurons in the central brain that perceive and interpret the courtship song. Nonetheless, these traits co-evolve such that females tend to prefer songs produced by males of their own species (*Bennet-Clark and Ewing, 1969*; *Ding et al., 2019*).

The analysis framework introduced here represents the first attempt to analyze full behavioral repertoires to gain insight into evolution. In principle, this approach could be applied to any data set where a large number of behaviors have been sampled in many species. However, there are several areas where one could add future improvements to this approach. First, we recorded behavior from only six species of flies. Adding additional species would place more constraints on the evolutionary dynamics, likely resulting in less variance in the ancestral state estimations and potentially adding more structure to the relatively low rank (i.e., highly modular) covariance matrices. Additionally, further work is required to determine the balance between sampling within and between strains and species that optimizes estimates of evolutionary dynamics.

Second, our framework assumes that all evolutionary changes in behavior resemble a diffusion process. Although this assumption is a reasonable initial hypothesis (*Felsenstein, 1985*), it may be possible to test this assumption. For example, deeper sampling of additional species may allow identification of specific behaviors on particular lineages where neutrality can be rejected (*Tajima, 1993*). If evidence emerges that the analyzed behaviors do not evolve under a diffusion process but under stabilizing selection, for example, the model for ancestral reconstruction can be changed from a Brownian motion to an Ornstein-Uhlenbeck process (*Martins and Hansen, 1997*; *Hansen and Martins, 1996*; *Royer-Carenzi and Didier, 2016*). Such a change can be implemented by altering the structure of the phylogenetic matrix, $A$ (*Martins and Hansen, 1997*; *Caetano and Beaulieu, 2020*), but without other alterations to the overall methodology presented here.

Another potential limitation of our analysis is that some of the observed inter-specific differences may reflect species-specific responses to environmental factors like room temperature or humidity, rather than underlying genetic or developmental factors. Two observations mitigate against this possibility, however. First, there were no significant differences in overall activity level of the different species, which would be a key indicator of environment-induced covariance. Second, the intra-species covariance matrix (derived from data from all species) agrees well with previous findings within a single species (*Berman et al., 2016*), implying that many of the potential environmental co-varying factors are shared across all six species.

In addition, all of our current analyses ignored the temporal structure of behavior and sequences of movements. While we found that the intra-specific variance has a similar structure to temporal structure that we reported previously (*Berman et al., 2016*), the order in which behaviors occur may also provide important biological information, especially during events like courtship or aggression. It should be possible to incorporate temporal structure directly into the regression (*Caetano and Beaulieu, 2020*). Deciding exactly which quantities to measure and how they should be incorporated, however, are complex questions that are outside the scope of this initial study. In addition, the number of fit parameters, already over 18,000 here, would need to grow even larger to accommodate modeling transition rates between the behaviors as traits themselves. Thus, fitting such models would necessitate even larger data sets than the one collected here. Moreover, because the Perron-Frobenius Theorem mathematically couples the transition probabilities between behaviors and the probabilities of the fly performing a given behavior, additional care (and data) is required to ensure that observed differences in behavior are due to changes in temporal structure rather than changes in the frequencies of performing a given behavior.

Lastly, capturing the full range of animal behaviors for a large number of animals presents a number of technological challenges, which is why we focused on measuring behavior in a highly simplified environment. However, a more complete understanding of the structure of behavior will require

more sophisticated ways to capture behavioral dynamics in more naturalistic settings and during complex social arrangements. While modern deep learning methods have made tracking animals in more realistic settings increasingly plausible (*Pereira et al., 2019*; *Mathis and Mathis, 2020*), there are still considerable hurdles to translating this information into a form that can be subjected to the kind of analysis we propose here.

Despite these limitations, this work represents a new way to quantitatively characterize the evolution of complex behaviors, which may provide new phenotypes that can be subjected to experimental analysis. In the absence of a behavioral fossil record, reconstructing ancestral behaviors requires an inferential approach like the one we present here. In addition, more complex models could be built to test assumptions of the diffusion-based model we employed. Finally, a strength of our approach is that it makes falsifiable predictions about how behaviors are linked mechanistically, providing predictions that can be tested experimentally to provide further insight into the genetic and neurobiological structure of behavior.

# Materials and methods

## Data collection

All fly handling and imaging of fly behavior followed the procedures described in *Cande et al., 2018*, excepting that we did not provide retinol-free food to any of the animals, nor we did provide any red light cycling during the experiments. Individual male flies were collected upon eclosion and housed singly in 2 mL wells in a 96-well 'condo,' with food deposited in the bottom of each well, which was sealed at the top with an airpore sheet. In total, we collected data from 561 individual from 18 strains and six species. Flies were imaged at age 7–12 days, within 4 hr of lights on. Individuals were sampled from multiple strains and species: three strains of *D. mauritiana* (*mau29*: 29 flies, *mau317*: 35 flies, *mau318*: 32 flies), four strains of *D. melanogaster* (*Canton-S*: 31 flies, *Oregon-R*: 33 flies, *mel54*: 34 flies, *mel56*: 31 flies), three strains of *D. santomea* (*san00*: 29 flies, *san1482*: 33 flies, *STO OBAT*: 22 flies), three strains of *D. sechellia* (*sech28*: 32 flies, *sech340*: 25 flies, *sech349*: 33 flies), three strains of *D. simulans* (*sim5*: 33 flies, *sim199*: 30 flies, *Oxnard*: 34 flies), and two strains of *D. yakuba* (*yak01*: 34 flies, *CYO2*: 31 flies).

## Generalized linear mixed model

We fit our GLMM (*Equation 1*) using the software introduced in *Hadfield, 2010*. The covariance matrices $V^{(e)}$ and $V^{(a)} \in \mathbb{R}^{K \times K}$, $K = 134$ and the mean vector $\vec{\mu} \in \mathbb{R}^{K \times 1}$ were inferred from the posterior distribution via MCMC sampling. Prior distributions for the covariance matrices were given by Inverse Wishart Distributions (conjugate priors for the multi-Gaussian model) with $K$ degrees of freedom and $\frac{1}{K+1}\frac{I+J}{2}$ as scale matrix, with $J$ and $I$ the unit and identity matrices respectively. Tree branch length were estimated from *Seetharam and Stuart, 2013*.

## Gelman-Rubin convergence diagnostic

This test evaluates MCMC convergence by analyzing the difference between several Markov chains. Specifically, we compare the estimated between-chains and within-chain variances for each parameter of the model. Large differences between these variances indicate non-convergence (*Gelman and Rubin, 1992*). Let $\theta$ be a model parameter of interest and $\{\theta_m\}_{t=1}^{N}$ be the $m$ th simulated chain, $m = 1, 2, ..., M$. Denote, $\hat{\theta}_m$ and $\hat{\sigma}_m^2$ be the sample posterior mean and variance of the $m$ th chain. If $\hat{\theta} = \frac{1}{M}\sum_{m=1}^{M}\hat{\theta}_m$ is the overall posterior mean estimator, the between-chains ($B$) and within-chain ($W$) variances are given by:

$$B = \frac{N}{M-1}\sum_{m=1}^{M}(\hat{\theta}_m - \hat{\theta})^2, W = \frac{1}{M}\sum_{m=1}^{M}\hat{\sigma}_m^2. \tag{5}$$

In *Gelman and Rubin, 1992*, it is shown that the following weighted average of $W$ and $B$ is an unbiased estimator of the marginal posterior variance of $\theta : \hat{V} = \frac{N-1}{N}W + \frac{M+1}{NM}B$. The ratio $\hat{V}/W$ should get close to one as the M chains converge to the target distribution with $N \to \infty$. In reference (*Brooks and Gelman, 1998*) this ratio known as the Potential Scale Reduction Factor (PSRF) was

corrected to account for the the sampling variability using $R_c = \sqrt{\frac{d+3}{d+1}\frac{\hat{V}}{W}}$, where $d$ is the degrees of freedom estimate of a t-distribution. Values of PSRF for all model parameters such that $R_c < 1.1$ are used in *Brooks and Gelman, 1998* as a criteria for convergence of the MCMC chains. Here, we used 20 independent chains, each with a different initialization.

### Deviance information criterion (DIC)

The DIC is used as a Bayesian model selection criteria in problems where there is hierarchical structure to the underlying models and where the correct effective number of parameters is difficult to ascertain (*Spiegelhalter et al., 2002*). These aspects are often found in models like ours where the posterior distributions have been sampled using Markov Chain Monte Carlo. The DIC is defined as follows:

$$
\begin{aligned}
\text{DIC} &= \overline{D(\theta)} + P_D, \quad \text{with} \\
\overline{D(\theta)} &= \mathbb{E}_{p(\theta|y)}[D(\theta)], \\
p_D &= \overline{D(\theta)} - \overline{D(\bar{\theta})}
\end{aligned}
\tag{6}
$$

where $D(\theta) = -2\log P(y \mid \theta) + 2\log f(y)$ is called the deviance ($f(y)$ denotes a function of the data alone and $P(y \mid \theta)$ corresponds to the likelihood of the model under evaluation). Hence, the posterior mean, $\overline{D(\theta)}$, can be considered as a Bayesian measure of fit. $P_D$ represents the effective number of parameters, where $D(\bar{\theta})$ is the deviance evaluated at the posterior mean of the parameters $\bar{\theta}$. Note that (i) both quantities needed to calculate DIC, $\overline{D(\theta)}$, and $D(\bar{\theta})$, can be readily estimated from the samples generated by MCMC, and (ii) alternatively, we can also re-write $DIC = D(\bar{\theta}) + 2P_D$. This is similar in form to the better-known Akaike Information criterion (AIC) (*Akaike, 1973*), for models with negligible prior information or for large data sets where the likelihood dominates over the prior.

### Comparing the focused trait and full repertoire models

To build a model where behavioral traits evolve independently from each other, we fit each a single trait GLMM for each behavior j:

$$
l_j = \mu_j + \vec{\rho}_j + \vec{e}_j,
\tag{7}
$$

where $l_j$ denotes the logarithm of the behavioral trait $P_j$, $\mu_j$ is the logarithm of the mean behavior of the common ancestor (treated as the fixed effect of this model), and $\vec{\rho}_j$ and $\vec{e}_j$ are the random effects corresponding to the phylogenetic and individual variability, respectively. Similar to the multi-response model, these random effects are normally distributed from $\mathcal{N}(\vec{0}, A \otimes \sigma_j)$ and $\mathcal{N}(\vec{0}, A \otimes \alpha_j)$ with $\sigma_j$ and $\alpha_j$ (single numbers) corresponding to the phylogenetic and individual inferred variances and A the phylogenetic matrix defined in the main text. Prior distributions for the variances are given by inverse-Wishart distributions with 1.002 degrees of freedom and scale parameter equal to the variance of the logarithm of the corresponding behavioral trait.

We fit these models using 10 bootstrapped data sets and obtained an average DIC value of $(230 \pm 2) \times 10^3$. Note that in the single-trait model, since each behavior is treated independently, the likelihood gets factorized in terms of the individual likelihoods corresponding to each behavioral trait: $P(l_1, l_2, ..., l_K \mid \vec{\theta}) = \prod_{i=1}^{K} P(l_i \mid \theta_i)$. Therefore, the DIC (estimated in terms of the log-likelihood) is given by $DIC = \sum_{i=1}^{K} DIC_i$, where $DIC_i$ is calculated for each single trait GLMM.

In contrast, the complete GLMM model (described in the main text and in the section above) had a significantly lower average DIC value of $(114 \pm 2) \times 10^3$ (calculated over 10 bootstrapped data sets as well).

### Information-based clustering

The information-based clustering approach used in this article (originally introduced in *Slonim et al., 2005*) minimizes the distance between elements within clusters, while also compressing the original representation as much as possible. More precisely, the method minimizes the functional

$$\mathcal{F} = \langle d \rangle + TI(C;i), \tag{8}$$

where $I(C;i) = \sum_{i=1}^{N} \sum_{C=1}^{N_c} P(C;i) \log\left[\frac{P(C|i)}{P(C)}\right]$ is the mutual information between the original behavioral variable i and the clustering $C.> = \sum_{C=1}^{N_c} P(C)d(C)$, and $d(C)$ is the average distance of elements chosen out of a single cluster:

$$d(C) = \sum_{i_1}^{N} \sum_{i_2}^{N} P(i_1 \mid C)P(i_2 \mid C)d(i_1, i_2), \tag{9}$$

with $d(i_1, i_2)$ being the distance measure between a pair of elements and $P(i \mid C)$ being the probability to find element $i$ in cluster $C.T$ is a Lagrange multiplier that modulates the relative importance of minimizing the average within-cluster distance and simplifying the clustering.

Given $|C| = N_c$, $T$ and a random initial condition for $P(C \mid i)$, a solution is obtained by iterating a set of self-consistent equations (**Slonim et al., 2005**) until the convergence criteria $\frac{\mathcal{F}_t - \mathcal{F}_{t+1}}{\mathcal{F}_t} < 10^{-5}$ is satisfied. We chose 40,000 different initial conditions for $P(C|i)$, along with randomly chosen values of $T \in [0.1, 1000]$ and $N_c \in \{2, 3, \ldots, 20\}$. For each set of initial conditions and parameters, we performed the optimization until the convergence criterion was met. We defined the Pareto front as the set of solutions $P(C \mid i)$ such that no other solution presents a smaller $\langle d \rangle$ and a smaller $I(C;i)$, and we only kept solutions that were along this front (eliminating duplicates). Finally, for each number of clusters we selected the solution with the lowest $\langle d \rangle$.

For each number of clusters, we assess the modularity of the found solution by comparing $\langle d \rangle$ for the solution to the average distance corresponding to random cluster assignments. These assignments are made in such a way that the amount of elements per cluster is conserved by randomly shuffling the vector that assigns each behavior to a particular cluster. The values presented in the main text correspond to the mean and standard deviation of $\langle d \rangle$ over 50 different random trials.

## Deterministic Information Bottleneck

We use the Deterministic Information Bottleneck (DIB) method (**Strouse and Schwab, 2017**) to find coarse-grainings of the behavioral space that optimally predict future states. Inspired by the Information Bottleneck (IB) (**Tishby et al., 1999**), given two measured variables, $X$ and $Y$, the DIB method finds a clustering, $Z$, of $X$, where $Z$ is maximally informative of $Y$, but is as simple as possible. Specifically, we minimize the functional:

$$\mathcal{J} = -I(Y;Z) + \gamma\mathcal{H}(Z) \tag{10}$$

with respect to $p(z \in Z | x \in X)$. Here, $\gamma$ is a Lagrange multiplier that modulates the relative importance of the two terms, with larger values of $\gamma$ resulting in simpler representations.

In practice, to compute this minimum for a given value of $\gamma$ and an initial condition for $p(z|x)$, we minimize

$$\mathcal{J}(\alpha) = \gamma H(Z) - \alpha H(Z \mid X) - I(Y;Z) \tag{11}$$

with respect to $p(z \in Z | x \in X)$ and take the limit as $\alpha \to 0$, following the self-consistent equation procedure described in **Strouse and Schwab, 2017**.

To apply DIB to the behavioral dynamics, we count time in units of the transitions between states, providing a discrete time series of behaviors: $b(n)$ can thus be one of $N = 134$ different integer values at each discrete time $n$. Here, we relate the joint distributions of $b(n)$ ($X$ in **Equation 10**) and $b(n + \tau)$ ($Y$) through a coarse-grained clustering of the behavioral states ($Z$). Similar to our approach with information-based clustering (see previous section), we chose 10,000 different pairs of random values for $\gamma$ between 0.1 and $10^4$ and $N_c$ between 2 and 30 clusters. Given $N_c$, $\gamma$ and a random initial condition for $p(t \mid x)$, we find a solution by iterating through a set of self-consistent equations (**Strouse and Schwab, 2017**) until the convergence criteria (an absolute change in the function of less than $10^{-6}$) is satisfied. If any cluster has its probability become zero at any iteration, then that cluster is dropped for all future iterations. Thus, $N_c$ is the maximum number of clusters that can be returned. Of these 10,000 solutions, we keep all solutions that are on the Pareto front (i.e., no other

solution has both a higher $I(Y;Z)$ and a smaller $H(Z)$). The displayed clusters are the solutions on the Pareto front with the largest $I(Y;Z)$ for a given number of clusters.

## Weighted similarity index

We quantify the similarity between clustering partitions using the Weighted Similarity Index (WSI), a modification of the Rand Index (*Rand, 1971*) such that behaviors contribute the index according to their overall probability. Specifically,

$$\mathrm{WSI} = \sum_{i,j \in S_a} W_{ij} + \sum_{k,l \in S_b} W_{kl}, \; W_{ij} = \frac{P_i P_k}{\sum_{kl} P_k P_l}, \tag{12}$$

where $S_a(S_b)$ is the set of pairs of behaviors that belong to the same (different) cluster in the two partitions and $P_k$ is the probability of observing behavior $k$.

## Acknowledgements

We thank Ilya Nemenman, Jennifer Rieser, and Daniel Weissman for their helpful comments on the manuscript. DGH was supported by Programa Raices from the MinCyT. CR was supported by the NSF Physics of Living Systems Student Research Network (1806833). GJB. was supported by NIMH R01 MH115831-01, the Human Frontier Science Program (RGY0076/2018), and a Cottrell Scholar Award, a program of the Research Corporation for Science Advancement (25999). JC, DLS, and GJB were supported by the Howard Hughes Medical Institute and the Janelia visiting researcher program.

## Additional information

### Competing interests

Gordon J Berman: Reviewing editor, *eLife*. The other authors declare that no competing interests exist.

### Funding

| Funder | Grant reference number | Author |
|---|---|---|
| National Institute of Mental Health | MH115831-01 | Gordon J Berman |
| Human Frontier Science Program | RGY0076/2018 | Gordon J Berman |
| Howard Hughes Medical Institute | | Jessica Cande David L Stern Gordon J Berman |
| Research Corporation for Science Advancement | 25999 | Gordon J Berman |
| National Science Foundation | 1806833 | Catalina Rivera |
| Ministerio de Ciencia, Tecnología e Innovación de Argentina | | Damián G Hernández |

The funders had no role in study design, data collection and interpretation, or the decision to submit the work for publication.

### Author contributions

Damián G Hernández, Catalina Rivera, Conceptualization, Data curation, Software, Formal analysis, Validation, Investigation, Visualization, Methodology, Writing - original draft, Writing - review and editing; Jessica Cande, Conceptualization, Data curation, Investigation, Methodology, Writing - review and editing; Baohua Zhou, Conceptualization, Software, Investigation, Methodology, Writing - review and editing; David L Stern, Conceptualization, Resources, Funding acquisition, Investigation,

Visualization, Methodology, Writing - review and editing; Gordon J Berman, Conceptualization, Resources, Data curation, Software, Formal analysis, Supervision, Funding acquisition, Investigation, Visualization, Methodology, Writing - original draft, Writing - review and editing

**Author ORCIDs**
Damián G Hernández http://orcid.org/0000-0002-8995-7495
David L Stern https://orcid.org/0000-0002-1847-6483
Gordon J Berman https://orcid.org/0000-0003-3588-7820

**Decision letter and Author response**
Decision letter https://doi.org/10.7554/eLife.61806.sa1
Author response https://doi.org/10.7554/eLife.61806.sa2

## Additional files

### Supplementary files
- Source data 1. Fly behavior source data.

- Transparent reporting form

### Data availability

All behavioral region information is submitted with the article and is posted on GitHub (https://github.com/bermanlabemory/behavioral-evolution, copy archived at https://archive.softwareheritage.org/swh:1:rev:b01a6e3a2c7da193f38631dfe925c65229494d74). The original video data are too large to post (tens of TB), but will be made available upon request.

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
