## [Decision Letter]

**Acceptance summary:**

Different animal species exhibit distinct behavioral repertoires, even in cases where common ancestors are relatively recent. How do repertoires acquire species-specificity during evolution? This manuscript provides new methods to examine behavioral patterns of different *Drosophila* species, and even to reconstruct potentially ancestral behavioral modes. Consistent with past work, the authors find that behaviors are inherited in clusters. Overall, this stimulating paper presents an exciting method for the unbiased study of behavioral evolution based on large datasets that are increasingly common in animal behavior, which differs from previous work focusing on specific traits.

**Decision letter after peer review:**

Thank you for submitting your article "A framework for studying behavioral evolution by reconstructing ancestral repertoires" for consideration by *eLife*. Your article has been reviewed by three peer reviewers, one of whom is a member of our Board of Reviewing Editors, and the evaluation has been overseen by Christian Rutz as the Senior Editor. The following individual involved in the review of your submission has agreed to reveal their identity: Iain D Couzin (Reviewer #2).

The reviewers have discussed their reviews with one another, and the Reviewing Editor has drafted this decision to help you prepare a revised submission.

As the editors have judged that your manuscript is of interest, but as described below that additional experiments are required before it is published, we would like to draw your attention to changes in our revision policy that we have made in response to COVID-19 (https://elifesciences.org/articles/57162). First, because many researchers have temporarily lost access to labs, we will give authors as much time as they need to submit revised manuscripts. We are also offering, if you choose, to post the manuscript to bioRxiv (if it is not already there) along with this decision letter and a formal designation that the manuscript is “in revision at eLife”. Please let us know if you would like to pursue this option. (If your work is more suitable for medRxiv, you will need to post the preprint yourself, as the mechanisms for us to do so are still in development.)

Summary:

In this paper, the authors build upon their previous (ground-breaking) work in which automated methodology was employed to quantify the structure of behaviour and its regulation in fruit flies. Specifically, here, they address whether such methodology can be insightful in the study of how behaviour evolves, again using fruit flies as convenient and powerful model species (their phylogenetic relationships are well-understood, they do well in the lab, and it has been previously shown how well the behaviour of these species can be decomposed using automated methodology). There are a host of reasons why such an analysis is of value, not least because revealing the hidden structure of behavioural regulation has the potential to be informative regarding ancestral behavioural repertoires, and thus how it evolves. Overall, there is consensus that this is a thought-provoking paper that provides a valuable starting point in the analysis of behavioural evolution via detailed quantitative ethology. With added consideration of the factors discussed below this work makes a very helpful and novel contribution.

Essential revisions:

1. Consider behavioural transitions

All three reviewers were confused as to why transitions between states were not considered. Given that the authors have previously shown their capacity to reconstruct the hierarchical nested structure of behavioural states, typically represented as a network with transition probabilities between states, which would seemingly be extremely informative about the evolution of behavioural modules, all reviewers were left wondering why they did not consider comparisons of these networks -- especially given the array of network analysis tools available. Organisms, as they are well aware, don't just jump randomly from one behaviour to another, but there is a hierarchical organisation by which one behaviour influences the probability that another is elicited, and so on. Why did the authors not do this? There may be very good reasons, but it will be important for the reader to be informed. This would seem to be the most direct way to address how behaviour is structured. It is not clear to the reviewers exactly how much extra work this would be -- or how the scope of the paper would be required to change if transitions were considered. Thus, while formal inclusion of transitions into the framework is strongly preferred for revision, it is not absolutely required. At the minimum, the authors need to clarify exactly why transitions were not considered as well as what may be gained from such an approach in future work.

2. Confirm utility of the approach, its discoveries, and how it compares to more focused inquiries into the evolution of behaviour

The work suffers from a lack of clarity over why/how the method is superior to the focused trait approach. The fact that behaviours co-evolve in suites is by itself not especially novel. The authors should bring to the forefront, if possible, exactly how and why this approach is superior to others. Ultimately, the paper is an argument for the utility of studying repertoires rather than specific behaviours. The main thrust the paper tries to make is that we learn something new by looking at the evolution of repertoires of behaviour rather than focused analyses of specific behaviours (really they are looking at time budgets and perhaps that's a bit of a disconnect from their message as well). But they put forth in Figure 3 a reconstruction that seems to generate a nonsense behavioural repertoire for the ancestral species. So the authors should address the validity of using the specific phylogenetic methods to reconstruct behaviours based on the repertoires. They can address this point in two ways:

2.1. Generate simulations of repertoire evolution on the known *Drosophila* phylogeny and then take the endpoint repertoires they have evolved and assess how well the reconstruction of the known ancestral state works. People do these sorts of studies regularly to assess phylogenetic reconstruction approaches and methods (e.g., Royer-Carenzi and Didier 2016, J. Theoretical Biol.). This might be done with a real or arbitrary repertoire as the starting point for the simulation. This is crucial since it appears the rest of their findings hinge on the ancestral reconstruction.

2.2. As a complement to 2.1, the authors might assess the reconstruction of the repertoires in comparison to ancestral trait reconstructions for specific behavioural traits that might be more conventionally measured, assuming this can be extracted from the existing data. Part of the argument the authors make is that by considering the repertoire we achieve a more complete understanding of behavioral evolution. The approach seems very promising, though it is difficult to assess how it compares to other approaches for measuring behaviour. However good or bad the reconstructions of ancestral behaviours are for the repertoires, how do they compare to a focused trait approach? Perhaps better? Perhaps worse? Unless a comparison is done it's hard to say. More broadly, the paper argues for the utility of this approach to studying behaviour but it is not clear how it compares to analyses that focus on specific elements of behaviour rather than time budget vectors.

3. Make the paper more accessible for a general audience

A major issue with this paper was its inaccessibility, which could, unfortunately, ultimately reduce its impact. Most readers are by now familiar with past work using clustering methods (e.g., Cande et al., 2018; Berman et al., 2014) -- but many of the analyses in this paper are novel, difficult to penetrate, and poorly introduced for a general audience. The authors should take a fine-grained comb to this paper to make each analysis as accessible as possible. Simple fixes could include more plain language around what each analysis is testing/achieving, as well as consistency in terminology. Specifically, the final sentence of each paragraph could state in plain terms what the preceding analysis has just demonstrated, ruled-in, or ruled out.

[Editors' note: further revisions were suggested prior to acceptance, as described below.]

Thank you for submitting your article "A framework for studying behavioral evolution by reconstructing ancestral repertoires" for consideration by *eLife*. Your article has been reviewed by three peer reviewers, including Jesse H Goldberg as the Reviewing Editor and Reviewer #1, and the evaluation has been overseen by Christian Rutz as the Senior Editor. The following individual involved in the review of your submission has agreed to reveal their identity: Iain D Couzin (Reviewer #2).

The reviewers have discussed their reviews with one another, and the Reviewing Editor has drafted this decision letter to help you prepare a revised submission.

Please address continued concerns from Reviewer #3, and please attempt to clarify key sections of the text as recommended by Reviewer #1.

1) Reviewer #3: The paper has improved in many respects but the revision failed to deal with what I see as the most glaring issue (and which I had raised in my previous review).

All the extant species show biases in behaviors as shown by the hotter/browner colors in their behavioral maps. This suggests that we should expect that any species of *Drosophila*, extant or extinct, would be highly likely to have biases behaviors as well. However, the ancestrally reconstructed behavioral repertoire is totally washed out – there is no clear biases in behavior in the ancestral graph that is estimated in figure 3. This suggests to me that the method potentially has flaws.

I had suggested that the authors start with a PDF from one of their species and then simulate its evolution along the known *Drosophila* phylogeny and then try to reconstruct that repertoire using ancestral state reconstruction. If that process ends up showing a similarly 'flat' PDF where everything is similarly low probability (i.e. a totally blue PDF as shown in figure 3) then there is some cause for concern. If it were to retrieve a PDF similar in structure to the one that initiated the simulation, then that would be very reassuring.

The author responses seem to say that is what they have done in using the MCMC method but where the ancestral reconstruction analysis of a simulated evolutionary trajectory is to be found in the paper is unclear to me. I searched 'simulat' in the text of the manuscript and cannot see where trait evolution was simulated and then used as the starting values to try to evaluate how closely the ancestral reconstruction gets to the known ancestral starting point.

There is a lot in this paper, so though I feel this issue is important to address, does not necessarily preclude it from being published in my view. Though I would like to see the authors address the 'flatness' and thus seemingly unrealistic (it would seem to me at least) ancestral state that they present in figure 3.

2) Reviewer #1: The writing is slightly improved, though some sections could still be improved by concluding paragraphs not just with a mathematical result but also what it means in plain terms. E.g. Lines 264-7 conclude with the finding that the covariance matrix has a 'far from random modular structure;' this does not explain how this result relates to the title of the section (Individual Variability and long timescale correlations).

3) Please note that *eLife* has recently adopted the STRANGE guidelines for animal behaviour research:

https://doi.org/10.1038/d41586-020-01751-5

https://reviewer.elifesciences.org/author-guide/journal-policies

In your revisions, please consider scope for sampling biases in your study and how these may limit the generalisability of your findings, and make declarations as necessary. A few sentences in the section "Data collection" (lines 462-470) may suffice.

---

## [Author Response]

Essential revisions:1. Consider behavioural transitionsAll three reviewers were confused as to why transitions between states were not considered. Given that the authors have previously shown their capacity to reconstruct the hierarchical nested structure of behavioural states, typically represented as a network with transition probabilities between states, which would seemingly be extremely informative about the evolution of behavioural modules, all reviewers were left wondering why they did not consider comparisons of these networks -- especially given the array of network analysis tools available. Organisms, as they are well aware, don't just jump randomly from one behaviour to another, but there is a hierarchical organisation by which one behaviour influences the probability that another is elicited, and so on. Why did the authors not do this? There may be very good reasons, but it will be important for the reader to be informed. This would seem to be the most direct way to address how behaviour is structured. It is not clear to the reviewers exactly how much extra work this would be -- or how the scope of the paper would be required to change if transitions were considered. Thus, while formal inclusion of transitions into the framework is strongly preferred for revision, it is not absolutely required. At the minimum, the authors need to clarify exactly why transitions were not considered as well as what may be gained from such an approach in future work.

We agree that studying the evolution of the transition structure between behaviors is an important subject, and we are currently developing methods to study it. The primary difficulty here is the number of parameters involved. Because we are fitting two covariance matrices and an ancestral mean, we need to fit at least (*N* + 1) ∗*N* + *N* parameters, where *N* is the number of traits being reconstructed. Currently, we study a system that has *N* = 134 traits, leading to 18,224 parameters that we must fit with the GLMM. If we were to add the transition probabilities (or transition rates), this would require another ≈*N*^2^ parameters. Thus, instead of fitting a system with ≈ 18,000 parameters, our model would now have 327,284,280 parameters. While we would likely be able to perform the computation, we would be unlikely to believe the results at this point because our data set size is insufficient (the amount of time needed to accurately sample a transition matrix is a power of two longer than the amount of time needed to accurately sample the behavioral frequencies).

In addition, these traits (the transition matrix values) have different units than the behaviors, and the transition probabilities mathematically depend on the behavioral frequencies, since the first eigenvalue of the transition matrix is (by definition) proportional to the average behavioral frequency usage. Thus, it becomes necessary to add additional regularizing or normalizing factors, further complicating the analysis in a manner that is beyond the scope of what we aimed to demonstrate here.

Despite these difficulties, although transitions are not included in our actual model, one of the main points that we have highlighted is the similarity between the hierarchical structure that explains long timescale transitions, which we demonstrated previously, and the variability we observe here between individuals of the same species.

Again, though, this is a problem that we find extremely interesting, and we are actively working on exploring how to solve these technical and representational questions. We had some text describing this as a future direction in our Discussion section, but we have now expanded this section to further point out the importance of this question and some of the current technical challenges surrounding the study of the evolution of behavioral transitions (lines 428-441).

2. Confirm utility of the approach, its discoveries, and how it compares to more focused inquiries into the evolution of behaviourThe work suffers from a lack of clarity over why/how the method is superior to the focused trait approach. The fact that behaviours co-evolve in suites is by itself not especially novel. The authors should bring to the forefront, if possible, exactly how and why this approach is superior to others. Ultimately, the paper is an argument for the utility of studying repertoires rather than specific behaviours. The main thrust the paper tries to make is that we learn something new by looking at the evolution of repertoires of behaviour rather than focused analyses of specific behaviours (really they are looking at time budgets and perhaps that's a bit of a disconnect from their message as well). But they put forth in Figure 3 a reconstruction that seems to generate a nonsense behavioural repertoire for the ancestral species. So the authors should address the validity of using the specific phylogenetic methods to reconstruct behaviours based on the repertoires. They can address this point in two ways:2.1. Generate simulations of repertoire evolution on the known *Drosophila* phylogeny and then take the endpoint repertoires they have evolved and assess how well the reconstruction of the known ancestral state works. People do these sorts of studies regularly to assess phylogenetic reconstruction approaches and methods (e.g., Royer-Carenzi and Didier 2016, J. Theoretical Biol.). This might be done with a real or arbitrary repertoire as the starting point for the simulation. This is crucial since it appears the rest of their findings hinge on the ancestral reconstruction.

This process is precisely what we are doing with our MCMC method (which was originally developed by Hadfield, as cited in the text). Specifically, given a known tree structure, we select a common ancestral mean behavior and two behavioral covariance matrix such that the likelihood of observing the measured behavioral distribution (including the covariances between the behaviors) is maximized. In addition to showing that our model has converged (Figure 3, Supplement 1), we also show that the predicted behavioral means at the endpoint species match the data (Figure 3, Supplement 2).

2.2. As a complement to 2.1, the authors might assess the reconstruction of the repertoires in comparison to ancestral trait reconstructions for specific behavioural traits that might be more conventionally measured, assuming this can be extracted from the existing data. Part of the argument the authors make is that by considering the repertoire we achieve a more complete understanding of behavioral evolution. The approach seems very promising, though it is difficult to assess how it compares to other approaches for measuring behaviour. However good or bad the reconstructions of ancestral behaviours are for the repertoires, how do they compare to a focused trait approach? Perhaps better? Perhaps worse? Unless a comparison is done it's hard to say. More broadly, the paper argues for the utility of this approach to studying behaviour but it is not clear how it compares to analyses that focus on specific elements of behaviour rather than time budget vectors.

To address these points, we have added an extra section in the main text, as well as in Materials and methods, that shows a comparison between our model and a simpler model where behaviors evolve independently (lines 214-229 and 491-524, as well as Figure 3, Supplement 3). In order to compare the performance of models with different complexity, we need a measure that not only takes into account how well the data is fit (i.e., maximizes likelihood), but that also penalizes the addition of extra parameters in more complex models (in this case, the off-diagonal terms of the two behavioral covariance matrices). For hierarchical models fit using MCMC methods (such as ours), the deviance information criterion (DIC) is the standard approach for model selection (Spiegelhalter et al., 2002). The DIC is a generalization of the Akaike criterion (see Materials and methods for additional details), and it is proportional to the negative log-likelihood evaluated at the mean of the posterior parameters, plus a penalty term proportional to the effective number of parameters in the model. The values of DIC for our model are much smaller than the models with a focused analysis, indicating that modeling covariance between behaviors explains the observed data much better than treating all behaviors independently. The implications of this result have been further expanded and discussed in the text, and we thank the reviewers for suggesting this validations approach.

3. Make the paper more accessible for a general audienceA major issue with this paper was its inaccessibility, which could, unfortunately, ultimately reduce its impact. Most readers are by now familiar with past work using clustering methods (e.g., Cande et al., 2018; Berman et al., 2014) -- but many of the analyses in this paper are novel, difficult to penetrate, and poorly introduced for a general audience. The authors should take a fine-grained comb to this paper to make each analysis as accessible as possible. Simple fixes could include more plain language around what each analysis is testing/achieving, as well as consistency in terminology. Specifically, the final sentence of each paragraph could state in plain terms what the preceding analysis has just demonstrated, ruled-in, or ruled out.

We agree with the reviewers that many aspects of the original manuscript text could have benefited from additional prose explaining the rationale and conclusions of the analysis. Following these comments, we have now included many additional clarifying comments to make the text more accessible.

[Editors' note: further revisions were suggested prior to acceptance, as described below.]

Please address continued concerns from Reviewer #3, and please attempt to clarify key sections of the text as recommended by Reviewer #1.(1) Reviewer #3: The paper has improved in many respects but the revision failed to deal with what I see as the most glaring issue (and which I had raised in my previous review).All the extant species show biases in behaviors as shown by the hotter/browner colors in their behavioral maps. This suggests that we should expect that any species of *Drosophila*, extant or extinct, would be highly likely to have biases behaviors as well. However, the ancestrally reconstructed behavioral repertoire is totally washed out – there is no clear biases in behavior in the ancestral graph that is estimated in figure 3. This suggests to me that the method potentially has flaws.I had suggested that the authors start with a PDF from one of their species and then simulate its evolution along the known *Drosophila* phylogeny and then try to reconstruct that repertoire using ancestral state reconstruction. If that process ends up showing a similarly 'flat' PDF where everything is similarly low probability (i.e. a totally blue PDF as shown in figure 3) then there is some cause for concern. If it were to retrieve a PDF similar in structure to the one that initiated the simulation, then that would be very reassuring.The author responses seem to say that is what they have done in using the MCMC method but where the ancestral reconstruction analysis of a simulated evolutionary trajectory is to be found in the paper is unclear to me. I searched 'simulat' in the text of the manuscript and cannot see where trait evolution was simulated and then used as the starting values to try to evaluate how closely the ancestral reconstruction gets to the known ancestral starting point.There is a lot in this paper, so though I feel this issue is important to address, does not necessarily preclude it from being published in my view. Though I would like to see the authors address the 'flatness' and thus seemingly unrealistic (it would seem to me at least) ancestral state that they present in figure 3.

We agree that the way in which we presented Figure 3 provides the impression that the ancestral repertoire is “washed out.” This is an effect of the way in which we plotted our data, and we thank the reviewer for pointing out this less-than-ideal visualization. The new version of the figure shows the ancestral distribution in non-logarithmic values, showing that there is indeed structure in this repertoire. We agree that it would be worrying if the distribution was indeed flat, but as can be seen, this isn’t the case here. All of the brown/hotter colors in the subsequent repertoires to the right are differences of the logs, which better shows the behavioral alterations along the phylogeny, as they are often subtle.

As to the proposed simulation, due to the reviewer’s excellent suggestion in the previous round of reviews, we added Figure 3—figure supplement 2, which shows that starting from the inferred ancestral state, the model’s mean predictions are in agreement with the data, providing a good double-check of our model’s self-consistency. In other words, when starting from the endpoints, we reconstruct the ancestral state (this is how the model is fit in the first place), and starting from the fit ancestral state, the model predicts, on average, the correct endpoint states. In addition, Figure 3 Figure supplement 3 (also emerging from the reviewers’ suggestions), shows that the model predicts the behavioral covariances well. We did not need to do simulations, per se, to show these results, since these values can be directly numerically calculated from the model (hence the lack of the word ’simulation’ connected to these efforts in the manuscript).

(2) Reviewer #1: The writing is slightly improved, though some sections could still be improved by concluding paragraphs not just with a mathematical result but also what it means in plain terms. E.g. Lines 264-7 conclude with the finding that the covariance matrix has a 'far from random modular structure;' this does not explain how this result relates to the title of the section (Individual Variability and long timescale correlations).

We thank the reviewer for their comments, and we have adapted several passages in the text to increase the clarity of our methodology and to make connections between the mathematical results and the according biological significance (see the latex-diff document for all changes).

(3) Please note that eLife has recently adopted the STRANGE guidelines for animal behaviour research:https://doi.org/10.1038/d41586-020-01751-5https://reviewer.elifesciences.org/author-guide/journal-policiesIn your revisions, please consider scope for sampling biases in your study and how these may limit the generalisability of your findings, and make declarations as necessary. A few sentences in the section "Data collection" (lines 462-470) may suffice.

We thank the reviewers for pointing this out (even as an *eLife* reviewing editor, GJB was embarrassingly unaware of these guidelines). Although detailed information about how the flies were handled/housed was previously published in Cande et al., *eLife*, 2018 (and was pointed to accordingly via a citation), we added relevant details here to the Materials and methods section to make the manuscript more self-contained and to make our experimental details more clear.

As to sampling biases, we have provided a detailed list of the specific strains used in our experimental – all of which are readily available through the UCSD or Bloomington fly stocks. While we have no evidence that our precise choice of species/strains within the *D. melanogaster* species subgroup should generate bias, we do acknowledge in the Discussion section that our phylogentic reconstruction is likely under-constrained, and that adding more species/strains will “place more constraints on the evolutionary dynamics, likely resulting in less variance in the ancestral state estimations and potentially adding more structure to the relatively low rank (i.e., highly modular) covariance matrices. Additionally, further work is required to determine the balance between sampling within and between strains and species that optimizes estimates of evolutionary dynamics.”